# Development of Air Ventilation Garments with Small Fan Panels to Improve Thermal Comfort

**Mengmeng Zhao [1,\*], Chuansi Gao [2] and Min Wang [3]**

1   College of Textile and Clothing, Shanghai University of Engineering Science, Shanghai 201620, China
2   Aerosol and Climate Laboratory, Division of Ergonomics and Aerosol Technology, Department of Design Sciences, Lund University, 221 00 Lund, Sweden; chuansi.gao@design.lth.se
3   College of Fashion and Art Design, Donghua University, Shanghai 200051, China; minmin1202@dhu.edu.cn
\*   Correspondence: mengmengzhao@126.com or mengmengzhao@sues.edu.cn; Tel.: +86-021-67791297

**Abstract:** Air ventilation garments (AVGs) are reported to be effective in improving thermal comfort in hot environments in previous research. The purpose of this study was to develop AVGs with small fan panels and examine their cooling performance. Three AVGs equipped with more, much smaller sized ventilation fans were developed, including FFV (ten small fans all located on the front body), BBV (ten small fans all located on the back body), and FBV (six small fans located on the front body and four small fans located on the back body). Another garment, without ventilation fans but with the same structure and textile material, was made as a reference garment (CON). The cooling performance of the four garments was examined through subject trials in a moderately hot environment of 32 °C and 60% relative humidity. Simulated office work with 70 min of sedentary activity was performed. The results showed that the physiological indexes of the mean body skin temperature, the mean torso skin temperature, and the heart rate in the three AVG scenarios were significantly lower than those in the CON condition ($p < 0.05$). Thermal sensation, thermal comfort, and wetness sensation were also improved when wearing the three AVGs ($p < 0.05$). No significant difference was displayed among the three AVGs on the whole body and the whole torso ($p > 0.05$) due to the similarity of the air velocity created by the fan panels. A significant difference was found on the local torso skin, with FFV significantly reducing the chest and the belly skin temperature, and BBV significantly reducing the scapula and the lower back skin temperature ($p < 0.05$). This study indicates that the AVGs with the small fan panels were effective in reducing heat strain and improving thermal comfort, and thus are recommended for use in hot environments.

**Keywords:** integrated electrical fans; cooling garment; skin temperature; thermal sensation

## 1. Introduction

The world has undergone more heatwaves and heat-related morbidity and mortality in recent years, even in Europe where the islands and coastlines normally bring cooler weather [1–3]. Various cooling strategies have been developed and proposed to combat heat stress during heatwaves for people working or exercising in hot environments [4,5]. Wearable personal cooling garments are one of these strategies. There are many types of personal cooling clothing, for instance, liquid cooling garments [6,7]; cooled air cooling garments [8]; phase change material (PCM) garments [9–11]; air ventilation garments [12–16]; and hybrid cooling garments with both PCMs and air ventilation cooling [17–20], or PCMs and liquid cooling [21].

Air ventilation garments (AVGs) are personal cooling garments equipped with small fans. AVGs enhance heat dissipation using fans to circulate the ambient air in the microclimate between the human body and the clothing. Heat strain can be relieved by enhancing convective and evaporative cooling [12]. Since they use ambient air and not cooled air, AVGs need no compressors or pumps to obtain cooling sources (normally in the form of

cooled air or cooled liquid). Hence, AVGs are lightweight, portable, low-cost, and feasible compared to other types of personal cooling garments.

During recent decades, many studies have been carried out to investigate the cooling performance of AVGs. Hadid et al. [13] and Chinevere et al. [14] investigated the cooling effect of AVGs in very hot environments (40 °C) and found that AVGs relieved heat strain. Zhao et al. [15] studied the cooling effects of AVGs worn at a slightly lower air temperature (38 °C) and found the AVGs were effective in reducing torso skin temperature. Another study [17] of the cooling performance of hybrid cooling garments with both air ventilation and PCMs in a hot environment (37 °C) showed they were effective in alleviating heat strain in simulated construction work. Ashtekar et al. [18] performed a field study in which the cooling effect of AVGs on construction workers was compared with wearing habitual clothing. It was found that the cooling garment provided an affordable way of alleviating the discomfort and physiological strain caused by summer environmental heat exposure. Apart from these human studies, investigations on the effect of AVGs on heat and mass transfer using mathematic modeling and simulations were also carried out. Sun and Jasper [22] developed a two-dimensional numerical model for the convective and evaporative heat loss of AVGs with different ventilation velocities. Choudhary et al. [23] developed a three-dimensional numerical model of a virtual thermal manikin wearing an AVG. They also performed thermal manikin tests and human trials to validate the model and found good agreement between the numerical results and the experimental results. Both studies suggested that the higher airflow rate of AVGs brought more heat loss and a greater cooling effect.

Previous studies of AVGs indicated that the cooling performance of AVGs is mainly determined by two factors: (1) the ventilation unit of the AVGs [24] and (2) the clothing design features; for instance, the clothing size, which determines the air gap between the clothing and the body [22,25], the ventilation openings, or holes or eyelets [26,27], and the fabric permeability of the garment [28–30]. The ventilation unit, a crucial component of an AVG, consists of ventilation fans, an electric power supply and electric wires, etc. In previous studies, two bigger fans (for instance, a fan diameter of 9.8 cm or 10 cm) were usually situated with one at the left waist and the other at the right waist to improve garment aesthetics and balance their weight [15–17,24–26]. Theoretically, the bigger the ventilation fans, the higher the air velocity that is produced; thus, a bigger cooling effect is achieved. However, large fans may bring problems, for instance, by increasing the clothing weight and producing uncomfortable contact between the body and the fans (this was detected in Zhao et al.'s studies [15,16]). Therefore, it is important to ensure a good balance between the size and number of ventilation fans. In the study of Sun and Jasper [22], an AVG with 4-8 small fans of diameters of 1–2 cm was proposed. The authors established a two-dimensional model to analyze the AVG heat transfer along the skin surface. However, the proposed concept was not made into garments and no further research was carried out to investigate the influence of the size and number of the fans on the cooling performance.

In this study, the size and number of the ventilation fans were the key design factors in the AVGs. Three AVGs equipped with a smaller size and more ventilation fans were designed and manufactured. One AVG had small fans all located in the front body (FFV), one AVG had small fans all located in the back body (BBV), and one AVG had some fans located in the front body and some located in the back body (FBV). In addition, one normal garment without ventilation unit but with the same style and structure was also made as a reference garment (CON). The cooling effect of the three AVGs, as well as the reference garment, was tested and examined by subject trials in a moderately hot environment of 32 °C, with 60% relative humidity (RH). The purpose of the study was to compare and evaluate the cooling performance of these AVGs with the small fan panel designs in the target thermal environment to improve the design of this type of personal cooling garments. It was hypothesized that AVGs equipped with small fan panels could reduce heat strain and enhance thermal comfort and provide a different cooling effect.

## 2. Materials and Methods

### 2.1. Materials

#### 2.1.1. The Textile Materials of the AVGs

Three AVGs and a reference garment with different ventilation units were designed and made. These garments were made of twill woven fabric with 10% cotton and 90% polyester blend. The detailed information on the textile materials is shown in Table 1.

**Table 1.** Information of the textile materials.

| Fabric Name | Fiber Content | Fabric Thickness (mm) | Fabric Weight (g/m²) | Fabric Permeability (mm/s) | Moisture Vapor Transmission (g/m²·h) |
|---|---|---|---|---|---|
| Twill woven | 10% cotton, 90% polyester | 0.21 | 117.3 | 190.1 | 10.3 |

#### 2.1.2. The Structure of the AVGs

These AVGs were jacket styles and had the same garment sizes as shown in Table 2. The sleeves were adjustable from long sleeves to short sleeves and vice versa by fastening the button with the strip on the cuff.

**Table 2.** Garment size of the AVGs (the unit is cm).

| Garment Length | Bust Circumference | Hip Circumference | Waist Circumference | Shoulder Width | Sleeve Length | Sleeve Width |
|---|---|---|---|---|---|---|
| 65 | 106 | 106 | 102 | 40 | 57 | 18 |

The AVGs were integrated with small fan panels. The fans were set in three different ways according to the theory that the human body has different thermal capacities and sweating rates in different body regions [31,32]. They were either all set in the front body of the garment (abbrev. FFV) or the back body of the garment (abbrev. BBV), or 6 smaller ones were set in the front and 4 bigger ones were set in the back (abbrev. FBV), or no fans were set (the reference garment, abbrev. CON), as shown in Table 3. The total weight of the four clothing samples was 381.4 g (FFV), 383.7 g (BBV), 399.0 g (FBV) and 211.6 g (CON), respectively. The slight weight difference between the three AVGs was due to the different lengths of the electric wires.

#### 2.1.3. The Ventilation Unit of the AVGs

The ventilation unit of the AVGs consisted of small fans, electric wires and battery box. The small fans included two sizes: 6 smaller ones and 4 bigger ones. The size of the smaller fans was 3 cm × 3 cm × 1 cm and their weight was 7.3 g each. The size of the bigger fans was 5 cm × 5 cm × 1 cm and their weight was 16.5 g each. These 10 fans were connected in parallel by electric wires and were sewn in the garment through the small holes on the fans (shown in Figure 1).

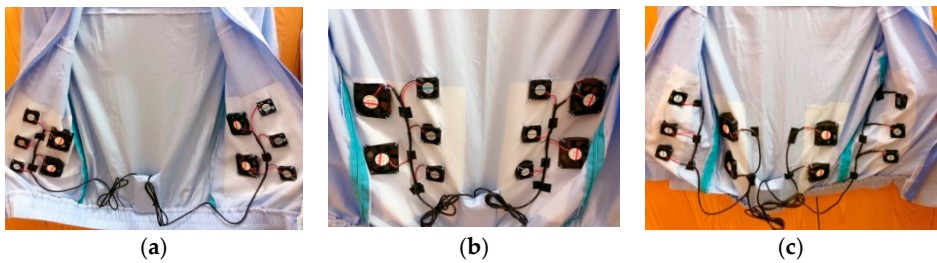

| (a) | (b) | (c) |
|---|---|---|

**Figure 1.** The AVGs with different fan panels. (**a**) FFV with small fans all set in the front body; (**b**) BBV with small fans all set in the back body; (**c**) FBV with small fans set both in the front and the back body.

The fans were powered by a portable battery (produced by ROMOSS Company, China). It had 10,000 mAh (37 Wh) capacity and could provide an electric power of 5 V. The weight of

the portable battery was 202 g and the size was 9.2 cm × 6.3 cm × 2.4 cm. The portable battery had two USB interfaces, which could connect the fans in two groups. Since the portable battery was small in size and light in weight, it was put in a pocket sewn at the bottom of the jacket. When the battery was fully charged, it could be used for 4.3 h for fan ventilation.

The air velocity provided by these AVGs was measured when the battery was fully charged. The AVGs were dressed on a female dummy (165/88) and were measured by a hot wire anemometer (Kanomax-6006, Kanomax, Kyoto, Japan). Five measuring points were marked at the front waist of the dummy, as shown in Figure 2. Each point was measured at least three times. The air velocity was calculated by the mean values of the five measuring points. The air velocity brought by these AVGs is shown in Table 3.

**Table 3.** Description of the AVGs.

| AVGs | Total Weight (g) | Air Velocity Measured at the One Side (m/s) | | Number of Fans | | Illustration of the AVGs | |
|------|------------------|---------|------|-------|------|------------|-----------|
| | | Front | Back | Front | Back | Front View | Back View |
| FFV | 381.4 | 1.23 | 0 | 10 | 0 | | |
| BBV | 383.7 | 0 | 1.22 | 0 | 10 | | |
| FBV | 399.0 | 0.77 | 0.80 | 6 | 4 | | |
| CON | 211.6 | 0 | 0 | 0 | 0 | | |

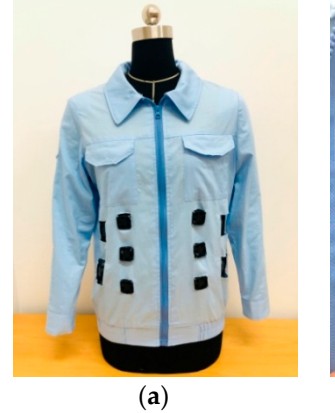
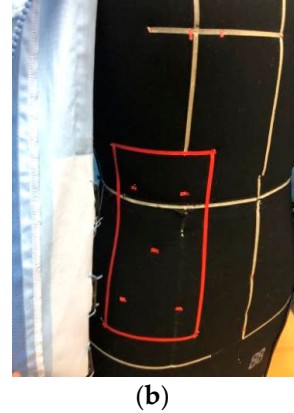

**(a)**　**(b)**

**Figure 2.** The measurement of the air velocity of the AVG. (**a**) the AVG dressed on the dummy; (**b**) the five measuring points on the dummy.

## 2.2. Experiment Methods

### 2.2.1. Subjects

Females are more susceptible to heat and more sensitive to change in environmental temperature [33,34]. Therefore, it was important to study the effect of cooling garments on female subjects. Ten female college students were recruited to participate in the test. They were chosen by the following criteria: (1) healthy and no chronic disease; (2) no smoking history; (3) no surgical history. Their age was (mean ± SD) 24 ± 1 years old, height was 161 ± 4 cm and weight was 55 ± 5 kg. Coffee and tea were not allowed 2 h before the test and heavy exercise was also not allowed 24 h before the test. The purpose and content of the experiment were briefly explained to the participants. Their written consent was obtained prior to the experiment. The study complied with the Declaration of Helsinki and it was approved by the University Ethical Committee.

### 2.2.2. Experiment Condition

Each subject participated in four tests with the four AVG cases: (1) wearing the AVG without small fans (the reference garment and no cooling condition, CON); (2) wearing the AVG with front ventilation cooling (FFV); (3) wearing the AVG with back ventilation cooling (BBV); and (4) wearing the AVG with both front and back ventilation cooling (FBV). A total of 40 tests (10 subjects × 4 scenarios) were performed. All the tests were randomized to diminish the effect of order. In all the tests, the climate chamber was set to 32 ± 0.5 °C, 60 ± 5% RH and 0.4 m/s air velocity.

### 2.2.3. Experiment Procedure

When the subjects came to the laboratory, they rested for several minutes. After that, they took off their outerwear and left their underwear on. Then, the experiment leaders put the sensors on the left side of the subject's body. The sensors were an ibutton wireless temperature logger (Maxim, CA, USA) with a diameter of 17.35 mm and thickness of 5.89 mm [35]. The ibutton sensors were set to measure the skin temperature at 15 s interval. The sensors were attached to the body by 3M medical tape at eight positions: the neck, the breast, the abdomen, the scapula, the lower back, the thigh, the shin and the hand. After the sensors were attached on the body, the subjects put on the AVG, long, causal pants, and sports shoes. Then, they sat on a chair, either reading a book or working with their laptops for 70 min (estimated metabolic rate was 1.2 Met according to the ISO 8996: 2021 standard [36]). The experiment scene was as shown in Figure 3. At the 20th min and the 45th min, the subjects could walk around in the chamber at a slow pace for 5 min. The experiment procedure is shown in Figure 4.

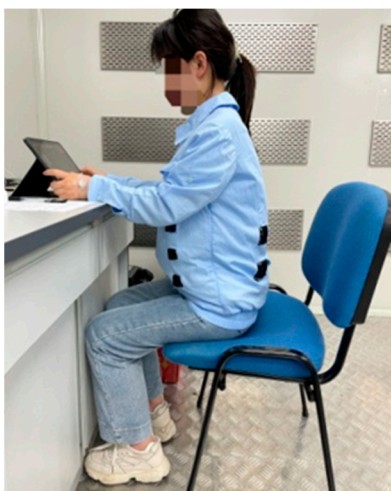

**Figure 3.** The test scene in the climate chamber.

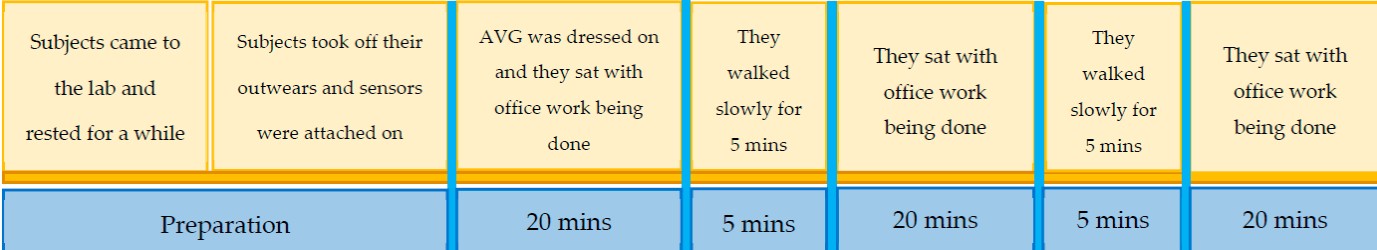

**Figure 4.** The experiment procedure.

During the 70 min test, heart rate was measured continuously by a wristband (Dido, Shenzhen, China). Ear canal temperature was also measured by an ear thermometer (Braun, IRT6520, Frankfurt, German) at 10 min interval. Perceptual responses, including thermal sensation, thermal comfort, the skin wetness sensation and the sensation of draught due to air ventilation, were asked and recorded at 10 min intervals. The thermal sensation was recorded by a 9-point scale, from −4 (very cold) to 4 (very hot) [37], and the thermal comfort was rated by a 7-point scale from −3 (very uncomfortable) to 3 (very comfortable) [38]. The skin wetness sensation was measured by a 4-point scale [39]. The draught sensation due to the fan ventilation was measured by a 5-point scale from −2 (unpleasant) to 2 (pleasant) [40]. These rating scales are described in Table 4.

**Table 4.** Rating scales for perceptual responses.

| Scale | Thermal Sensation | Thermal Comfort | Skin Wetness Sensation | Draught Sensation |
|---|---|---|---|---|
| −4 | Very cold | - | - | - |
| −3 | Cold | Very uncomfortable | - | - |
| −2 | Cool | Uncomfortable | - | Unpleasant |
| −1 | Slightly cool | Slightly uncomfortable | - | Slightly unpleasant |
| 0 | Neutral | Neutral | Neutral | Neutral |
| 1 | Slightly warm | Slightly comfortable | Slightly wet | Slightly pleasant |
| 2 | Warm | Comfortable | Wet | Pleasant |
| 3 | Hot | Very comfortable | Very wet | - |
| 4 | Very hot | - | - | - |

### 2.3. Calculation and Analysis

The mean skin temperature of the body ($T_{sk}$) was calculated by Equation [41]:

$$T_{sk} = 0.28T_{neck} + 0.28T_{scapula} + 0.16T_{hand} + 0.28T_{shin}$$

where $T_{sk}$ is the mean skin temperature of the body, °C. $T_{neck}$, $T_{scapula}$, $T_{hand}$ and $T_{shin}$ denote the skin temperature of the neck, the scapula, the hand and the shin, respectively, °C.

The mean skin temperature of the torso ($T_{torso}$) was calculated as [42]:

$$T_{torso} = 0.25T_{chest} + 0.25T_{scapula} + 0.25T_{abdomen} + 0.25T_{lower-back}$$

where $T_{torso}$ is the mean skin temperature of the torso, °C. $T_{chest}$, $T_{scapula}$, $T_{abdomen}$, and $T_{lower-back}$ denote the skin temperature of the chest, the scapula, the abdomen and the lower back, respectively, °C.

The presented data show mean values and standard deviation. The values at the 10th, 20th, 25th, 30th, 40th, 50th, 60th and 70th min were used to perform statistical analysis. A two-way ANOVA with repeated measures was employed (Time×Clothing) using SPSS 24.0 software (IBM Corp., Armonk, NY, USA) to evaluate the statistical differences in physiological response and the subjective ratings. A *p*-value of 0.05 was set to determine statistical difference. When the analysis revealed a significant difference, an LSD post hoc analysis was used to compare the four different clothing cases.

## 3. Results

### 3.1. Physiological Responses

3.1.1. Mean Skin Temperature

Figure 5 illustrates the evolution of the mean skin temperature over time, including the mean skin temperature of the body and the mean skin temperature of the torso. The mean skin temperature of the body increased with the evolution of time. At the 20th and 45th min, it decreased for 5 min and then increased till the end of the experiment. At the 50th min, it was significantly lower than those at other time points ($p < 0.05$). The mean skin temperature of the body in CON was the highest and it was the lowest in BBV among the four test cases. At the end of the test, the mean skin temperature of the body in the four AVG cases was 35.4 °C (CON), 35.1 °C (FFV), 35.0 °C (BBV) and 34.9 °C (FBV), respectively. Statistical analysis showed that the three AVGs had a significant cooling effect on the mean skin temperature compared to CON ($p < 0.05$). The three AVGs showed no significant difference in the mean skin temperature compared with one another ($p > 0.05$).

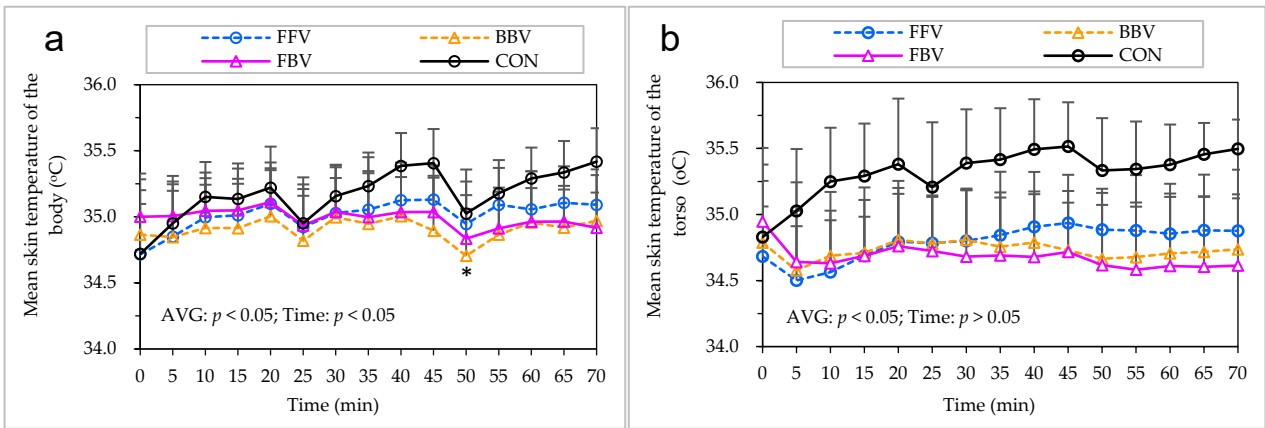

**Figure 5.** The mean skin temperature's evolution over time. (**a**) The mean skin temperature of the body; (**b**) the mean skin temperature of the torso; * means significant difference at $p < 0.05$ level.

The mean torso skin temperature in CON was significantly higher than those in the other three AVG cases ($p < 0.05$). The mean torso skin temperature in FBV was the lowest from the 15th to the 70th min. The reason for this might be that both the front and the back torso were integrated with small fans, which led to the largest skin area being cooled. At the end of the test, the mean torso skin temperatures in the four clothing cases were 35.5 °C (CON), 34.9 °C (FFV), 34.7 °C (BBV) and 34.6 °C (FBV), respectively. Statistical analysis showed that the three AVGs had a significant cooling effect on the mean torso skin temperature compared to CON ($p < 0.05$), but no significant difference was found among them regarding the mean torso skin temperature ($p > 0.05$).

3.1.2. Core Temperature and Heart Rate

Figure 6 shows the evolution of the ear canal temperature and the heart rate in real time. Ear canal temperature was around 37.1 °C during the 70 min test and no significant difference was found among the four AVG cases ($p > 0.05$). Heart rate in CON was the highest among the four AVG cases and showed significant difference between them ($p < 0.05$). At the 30th min, heart rate for the four cases was: 80.2 bmp (BBV), 83.0 bmp (FBV), 84.1 bmp (FFV) and 87.1 (CON). At the 70th min, heart rate for the four cases was: 79.1 (FBV), 80.9 (FFV), 82.4 (BBV) and 84.6 (CON).

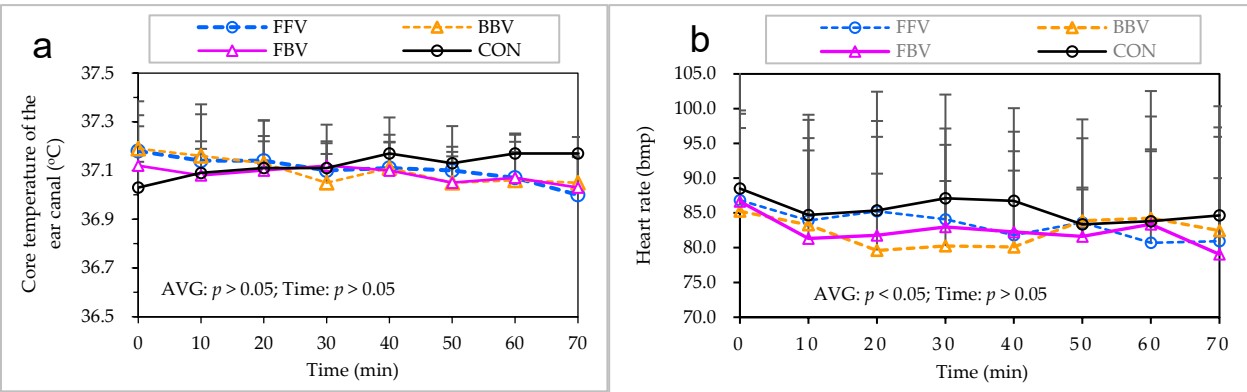

**Figure 6.** The evolution of the ear canal temperature and the heart rate. (**a**) The core temperature of the ear canal; (**b**) the heart rate.

### 3.2. Perceptual Responses

#### 3.2.1. Thermal Sensation

Figure 7 depicts the thermal sensation evolution of the whole body and the torso. The thermal sensation of the whole body in CON gradually increased over time. In comparison, thermal sensations of the whole body in the three AVG cases were between 0 (neural) and 1.2 (slightly warm) during the entire heat-exposure period. This indicated that the use of AVGs greatly reduced the thermal sensations of the whole body. Likewise, the thermal sensations of the torso in CON increased with the time and were much lower in the other three AVG cases. The thermal sensation of the torso was between 0 and 1 in BBV and FBV. It was lower in FFV from the 10th to the 40th min, showing a "slightly cool" sensation during this time period. At the end of the test, the thermal sensation of the torso in FFV was 0.4 (slightly cool), the lowest among the four test conditions. Statistical analysis showed that the use of AVGs had a significant cooling effect on the thermal sensation of both the whole body ($p < 0.05$) and the torso ($p < 0.05$). Likewise, no significant difference was found between the three AVGs in the thermal comfort of the whole body ($p > 0.05$) or on the torso ($p > 0.05$).

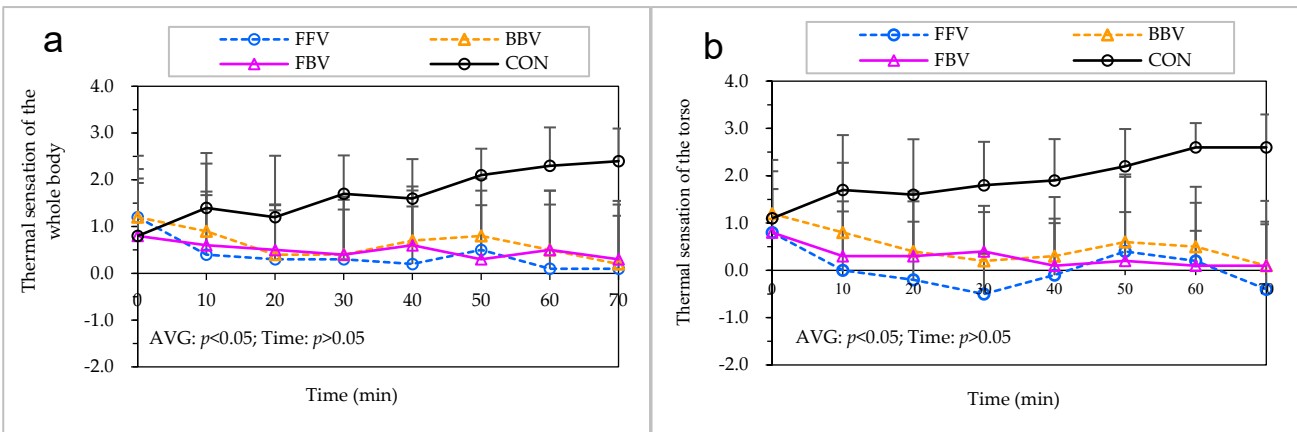

**Figure 7.** The thermal sensation over time. (**a**) The thermal sensation of the whole body; (**b**) the thermal sensation of the torso.

#### 3.2.2. Thermal Comfort

Figure 8 describes the thermal comfort evolution of the whole body and the torso. The thermal comfort of the whole body and the torso in CON decreased with time. At the end of the test, the values of the thermal comfort votes in CON were approximately −2, which indicated that the subjects felt "uncomfortable". In contrast, when the AVGs were worn, the thermal comfort of both the whole body and the torso were much higher

than those in CON conditions. The thermal comfort of the whole body and the torso in the three AVGs ranged between 0 (neutral) and 1 (slightly comfortable). The three AVGs had a significant cooling effect on the thermal comfort of both the whole body ($p < 0.05$) and the torso ($p < 0.05$) compared to CON, while no significant difference was found among the three AVGs in terms of thermal comfort ($p > 0.05$).

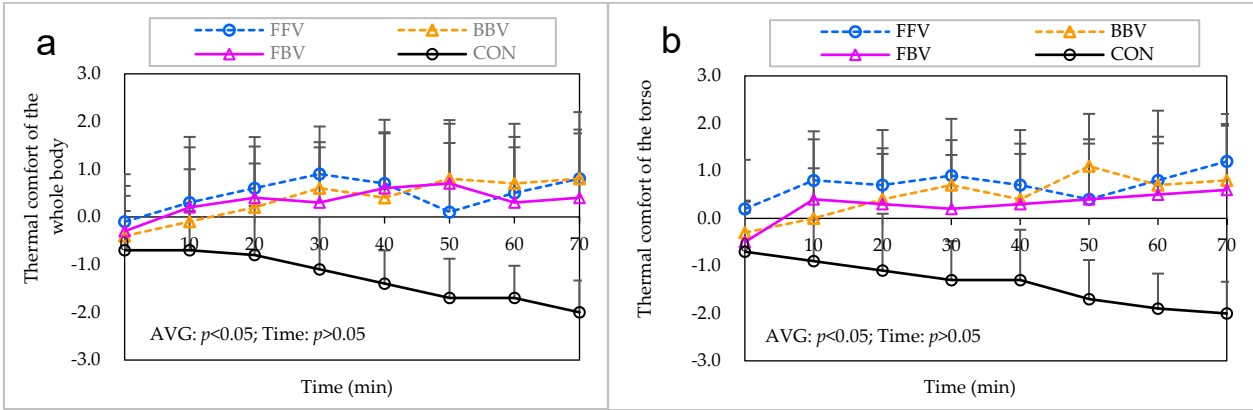

**Figure 8.** The thermal comfort over time. (**a**) The thermal comfort of the whole body; (**b**) the thermal comfort of the torso.

### 3.2.3. Skin Wetness Sensation

Figure 9 illustrates the wetness sensation experienced over the whole body and the torso over time. As shown in the figure, the skin wetness sensation experienced by the whole body and the torso in CON increased over time. At the end of the test, the values of the skin wetness sensation of the whole body and the torso were 1.6 (between "slightly wet" and "wet") and 1.7 (between "slightly wet" and "wet"), respectively. However, when the AVGs were put on during the test, the skin wetness sensation of the whole body and the torso were much lower. At the end of the test, the skin wetness sensation of the whole body was 0.3 (BBV), −0.3 (FBV) and −0.3 (FFV), and the skin wetness sensation of the torso was 0.3 (BBV), −0.2 (FBV) and −0.2 (FFV). Compared to CON, all three AVGs significantly reduced the skin wetness sensation of both the whole body ($p < 0.05$) and the torso ($p < 0.05$). The effects of the three AVGs on the skin wetness sensation were not significantly different ($p > 0.05$).

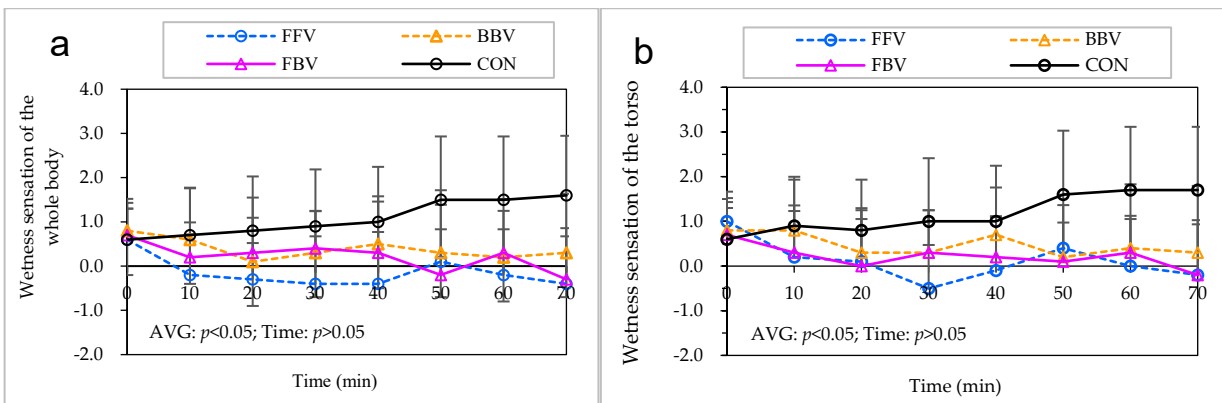

**Figure 9.** The wetness sensation over time. (**a**) The wetness sensation of the whole body; (**b**) the wetness sensation of the torso.

### 3.2.4. Draught Sensation

Draught sensations due to the ventilation produced by the small fans were also recorded and shown in Figure 10. Since the CON garments were not equipped with small

fans, the draught sensation felt in CON garments was not asked about. As shown in the figure, the draught sensation experienced by the whole body was between 0.3 (more than "neutral" and less than "slightly pleasant") and 1.2 (more than "slightly pleasant"). The draught sensation of the torso was between 0.2 (more than "neutral" and less than "slightly pleasant") and 1.2 (more than "slightly pleasant"). Statistical analysis showed that no significant difference was found among the three AVGs in terms of draught sensation for the whole body ($p > 0.05$) or for the torso ($p > 0.05$).

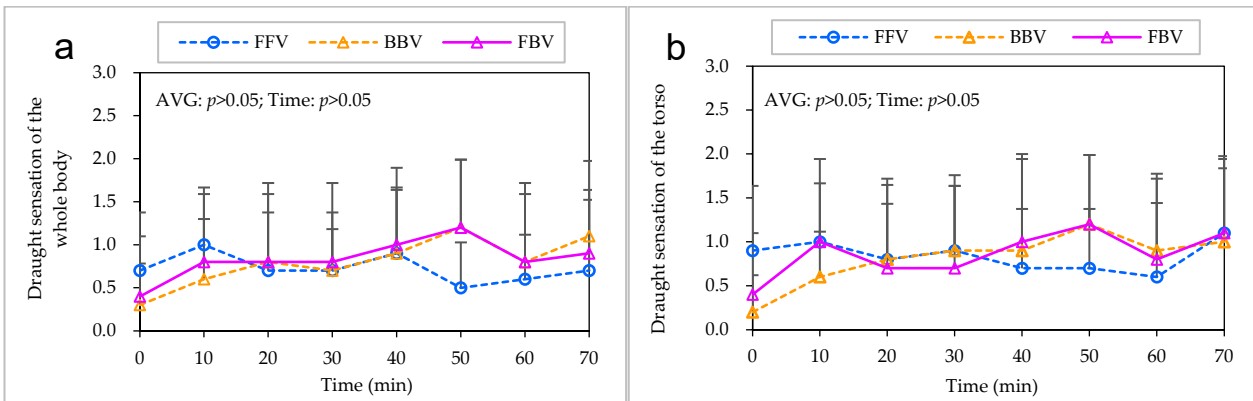

**Figure 10.** The draught sensation over time. (**a**) The draught sensation of the whole body; (**b**) the draught sensation of the torso.

### 3.3. Local Skin Temperature

Figure 11 displays the local skin temperature of the eight measuring points at the 25th, 50th and 70th min. The chest skin temperature was the highest for CON garments compared with the other three AVG cases; the same occurred for the belly, the scapula and the lower back skin temperature. This indicates that the small fan panels reduced the torso's local skin temperature. By comparison, the other four local skin temperatures of the neck, the hand, the thigh and the calf were not significantly reduced compared with that in CON except for the neck skin temperature at the 70th min.

For the four local torso skin temperatures of the chest, the belly, the scapula and the lower back, significant differences were found among the AVG cases. FFV significantly reduced chest skin temperature at the 25th, the 50th and the 70th min compared with the CON condition ($p < 0.05$). At the 50th and 70th min, FBV also significantly reduced this ($p < 0.05$). For the belly skin temperature, FFV showed a significant difference to BBV and CON at the 25th min ($p < 0.05$). FFV and FBV showed a significant difference to CON at the 50th and 70th min ($p < 0.05$). This indicated that FFV and FBV had small fans in the front torso and they significantly lowered the chest and belly skin temperature.

For the scapula skin temperature, BBV showed significant differences compared with CON at all three time points ($p < 0.05$). At the 50th and 70th min, BBV also displayed a significant difference with FFV ($p < 0.05$). For the lower back skin temperature, BBV displayed a significant difference compared with CON at all three time points ($p < 0.05$). At the 25th and the 70th min, FBV showed a significant difference compared with CON ($p < 0.05$). FBV also showed a significant difference compared with FFV at the 50th and 70th min ($p < 0.05$). Likewise, BBV and FBV had small fans in the back torso and they significantly lowered the scapula and lower back skin temperature. These observations indicated that the small fan panels had a significant cooling effect on the local torso skin temperature. However, in contrast, the cooling effect was not greatly obvious on other local skin temperatures of the neck, the hand, the thigh and the calf. For the neck skin temperature, a significant difference was found at the 70th min between the three AVGs and CON ($p < 0.05$). The reason for this might be that the neck skin also benefited and was cooled by the ventilated air, since the neck opening was one air outlet. For the hand skin temperature, FBV showed a significant difference compared with CON at the 70th

min ($p < 0.05$). For the calf skin temperature, FBV also showed a significant difference compared with CON at the 70th min ($p < 0.05$). This could be explained by FBV leading to the lowest torso skin temperature (see in Figure 5b), which might, in turn, reduce the local skin temperature of the limbs. For the thigh skin temperature, FFV showed a significant difference compared with CON at the 70th min ($p < 0.05$). Since the subjects were in a sedentary posture, the small fans in FFV were near the front thigh and made the thigh skin temperature lower too.

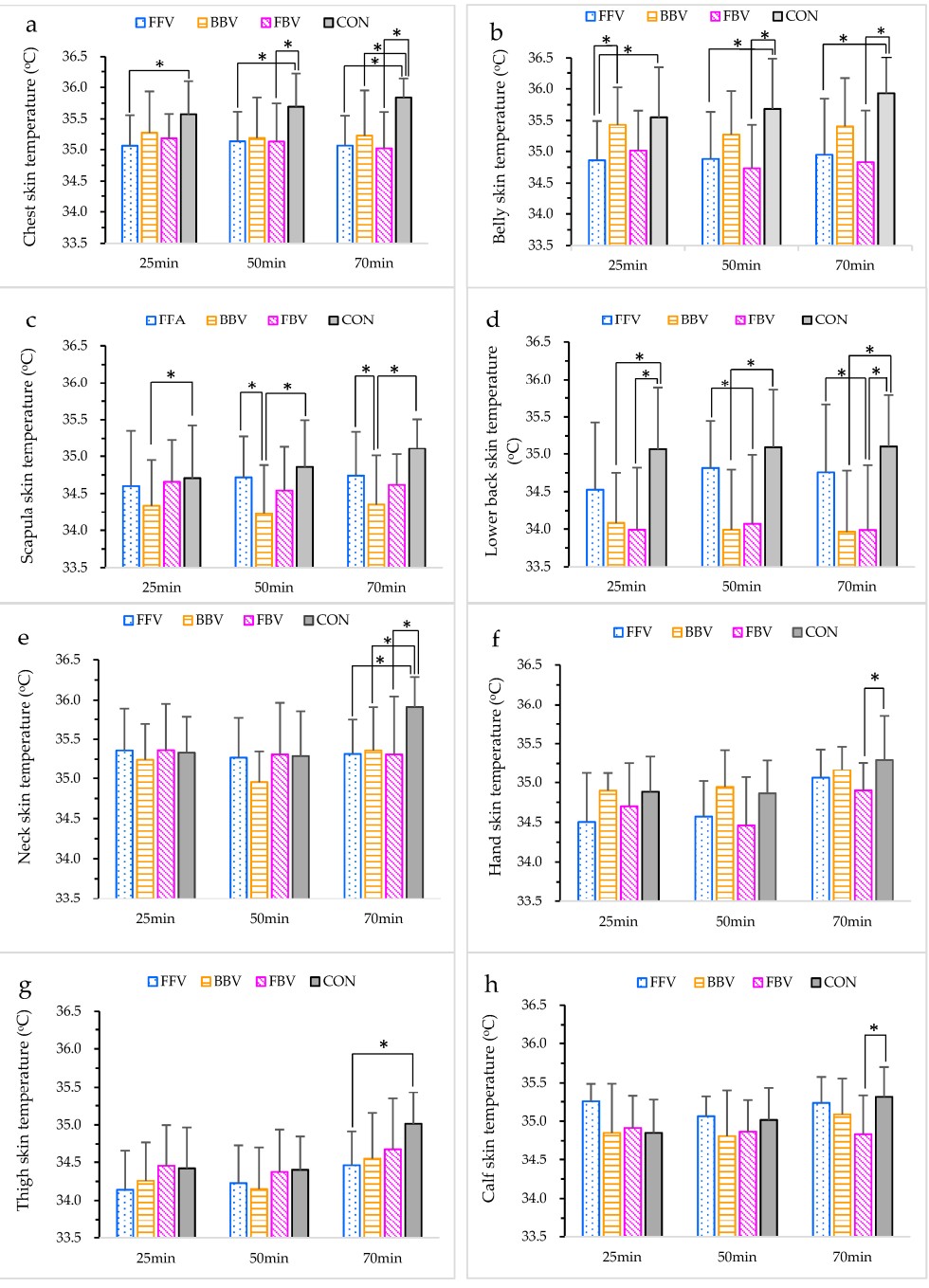

**Figure 11.** Local skin temperature at the 25th, 50th and 70th min. (**a**) Chest skin temperature; (**b**), belly skin temperature; (**c**), scapula skin temperature; (**d**), lower back skin temperature; (**e**) neck skin temperature; (**f**), hand skin temperature; (**g**), thigh skin temperature; (**h**), calf skin temperature. "*" means significant difference at $p < 0.05$ level.

## 4. Discussion

### 4.1. The Special Design of the Small Fan Panels

The AVGs or the hybrid cooling garments in previous studies were equipped with much bigger fans to achieve the goal of effective cooling. The size of the fans in those studies was 9.8 cm or 10 cm in diameter, and two such fans were equipped at the lower back of the garment [15–17,24–26]. The total weight of the AVGs was 0.6 kg in the studies of Zhao et al. [15,16,26] and 1.26 kg for a hybrid cooling garment in another study [17]. Those large ventilation fans were much heavier and increased the clothing weight for the wearer. In contrast, the total weight of the AVGs in the present study was less than 0.4 kg (shown in Table 3). The small fan panels were much smaller in size and much lighter in weight, and could reduce the unevenness imposed on the fabric due to the added weight.

The different arrangement of the small fan panels was designed based on the theory that the human body has different sweat rate and sweat production at different local regions. According to the studies of Havenith et al. [31] and Machado-Moreira et al. [32], the human torso has a non-uniform sweat secretion. The lower back of females has a higher sweat rate than the scapula, the chest and the belly. Therefore, the small fan panels were arranged at different torso regions to achieve the best cooling effect. The results of the local skin temperature in Figure 11 indicated that the different arrangement of the fan panels had a significant effect on the local torso skin temperature. For the chest and the belly skin temperature, FFV showed a significant difference as the fan panels were located at the front. For the scapula and the lower back skin temperature, BBV showed a significant difference as the fan panels were located at the back. FBV haa a significant effect on both the front torso skin and the back skin, i.e., the chest, the belly and the lower back skin temperature. These results confirmed those of a previous study [15] showing that the arrangement of the small fans had significant cooling effect on the local skin where the fans were located.

The three AVGs led to a slight difference in air velocity, as shown in Table 3. The air velocities provided by FFV and BBV were 1.23 m/s and 1.22 m/s, respectively. For FBV, air velocity was 0.77 m/s at the front body and 0.80 m/s at the back body. The physiological indexes of the mean body skin temperature, the mean torso skin temperature, the ear canal temperature and the heat rate displayed no significant difference between the three AVGs ($p > 0.05$) in terms of either the perceptual responses of the thermal sensation, the thermal comfort or the wetness sensation ($p > 0.05$). This indicates that the three AVGs with the different fan panels led to no significant difference to the cooling effect on the whole body (or the whole torso) due to the similarity in air velocity. A significant cooling effect was experienced for the local torso skin where the fans were located. This result also validated the mathematical modeling of the heat transfer of the AVGs conducted by former studies, showing that a higher air velocity leads to greater heat dissipation and, thus, a greater cooling effect [22,23]. Nevertheless, from the perspective of clothing weight balance and symmetrical aesthetic, the FBV with small fans both on the front and the back might be the best choice as there was no significant difference in the whole cooling effect.

### 4.2. The Cooling Effect of the AVGs

The results of the significantly lower skin temperature, lower heart rate and lower thermal sensation brought by the AVGs compared with the reference garment (without ventilation cooling, $p < 0.05$) indicate that the three AVGs reduced heat strain and improved thermal comfort. This confirmed that the AVGs developed with small fan panels are effective in cooling the body in a moderately hot environment. This was in line with a previous study [16] in which an AVG was used in a similar moderately hot environment (32 °C, 50% RH) for female subjects during post-exercise cooling. Two big fans (fan diameter was 10 cm) were equipped in that AVG, which differentiated it from the present study.

The draught sensation produced by the three AVGs was also rated and recorded. The rated draught sensation of the torso was between 0.2 (more than "neutral" and less than "slightly pleasant") and 1.2 (more than "slightly pleasant"). This hints that the fan panels of the three AVGs did not lead to much unpleasant draught sensation. In

contrast, previous studies of the desktop personalized ventilation air terminal device and desk fans demonstrate that these cooling strategies with air ventilation caused draught discomfort [43,44]. The small fan panels in this study were placed on the torso and the ventilated air moved along the trunk-clothing microclimate. Compared with a single bigger fan or two bigger fans (for instance, the fans with 9.8cm or 10 cm diameter in the aforementioned studies), the 10 small fans made more evenly distributed the air movement around the body.

In addition, the textile fabrics of common garments without a cooling source probably cannot have a similar magnitude of cooling power as the enhanced convective and evaporative cooling by fans; for a longer period, such as an 8 h work shift in a hot environment, cooling garments with small fans are needed to provide an extra cooling source. Therefore, the presented AVGs were effective in alleviating heat strain and improving thermal comfort, and are recommended in moderately hot environments in workplaces and offices in order to save energy, avoid cooling whole buildings, and promote sustainable development.

### *4.3. The Moderately Hot Environment*

The tests were performed in a moderately hot environment of 32 °C and 60% RH. The ventilated air of the AVGs was ambient air in which heat dissipation from the body mainly occurred by means of evaporation and convection. The forced air created by the small fan panels moved around the torso and enhanced evaporation and convection. Since it the ambient air was ventilated, not the cooled air, the AVGs did not need a compressor to produce cooled air. This ensured that the AVGs were lightweight, portable and affordable.

When the ambient temperature is around 35 °C or higher, the only heat loss avenue is through evaporation. In such a hot environment, the AVGs can still enhance evaporative cooling by accelerating sweat evaporation, as indicated by previous studies [13–15,45].

### *4.4. The Simulated Office Work*

It should be noted that the study simulated sedentary office work in a moderately hot environment. The participants either read a book or worked on a laptop. During the whole 70 min test period, they could walk slowly for several minutes around their chairs in the chamber. Thus, the metabolic heat produced by body was low (the estimated metabolic rate was 1.2 Mets). At higher activity levels in hotter environments, the cooling effect of the ventilation clothing may be different because of the higher internal and external heat load for the body. The study of Itani et al. [18] indicates that at different ambient conditions and human metabolic rates, personal cooling garments worked differently. The cooling effect that was produced was different. Therefore, personal cooling garments should be chosen according to the target working environment and working activity. The present study indicated that AVGs with a small fan panel are effective and recommended for simulated office work.

### 5. Conclusions

In this study, three AVGs with small fan panels placed at different torso positions were developed and their cooling performance was compared with a reference garment during simulated office work in a moderately hot environment. The results revealed that the three AVGs significantly reduced heat strain and improved thermal comfort compared with the reference garment sample, but no significant difference in cooling effect was found among the three AVGs on the whole body or the whole torso. This might be due to the similar air velocity produced by the fan panels. A significant difference was found on the local torso skin temperature, with FFV significantly reducing the front body skin temperature and BBV significantly reducing the back skin temperature. The AVGs with the small fan panels were much lighter and made the air movement along the torso-clothing microclimate more even; therefore, they are recommended for use in simulated office work. Furthermore, the heat production of the subjects was low while they remained in a sedentary posture during

the test period, so higher-intensity work or exercise should be performed to examine the cooling effect of the AVGs in future studies.

**Author Contributions:** Conceptualization, methodology, formal analysis, and writing by M.Z.; review and editing by C.G. and M.W. All authors have read and agreed to the published version of the manuscript.

**Funding:** This research was founded by the open fund of Key Laboratory of Clothing Design and Technology (Donghua University), Ministry of Education, China (No. KLCDT2020-06).

**Institutional Review Board Statement:** The study was conducted in accordance with the Declaration of Helsinki, and was approved by the Ethnical Committee of Shanghai University of Engineering Science (Approval no. EST-2023-014).

**Informed Consent Statement:** Informed consent was obtained from all subjects involved in the study.

**Data Availability Statement:** Data are available upon request from the corresponding author.

**Conflicts of Interest:** The authors declare no conflict of interest.

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
