# Peer review of "Development of Air Ventilation Garments with Small Fan Panels to Improve Thermal Comfort"

_sustainability, doi:10.3390/su15118452_

Round 1
Reviewer 1 Report (Previous Reviewer 1)
I would like to thank the authors for improving the manuscript.
Author Response
Dear reviewer,
Thank you very much for your review and suggestion.
Reviewer 2 Report (New Reviewer)
In general, very interesting work is presented. The aim of this research was to create a cooling performance assessment for AVGs that utilize smaller fan panels. To achieve this goal, three AVGs were developed, each equipped with smaller yet more numerous ventilation fans. These garments were labelled as FFV, BBV, and FBV, with each having ten small fans on either the front, back, or both sides of the garment, respectively. As a reference point, a garment without ventilation fans but made from the same material and structure was also created (CON). Subject trials were conducted in a moderately hot environment, where 70 minutes of sedentary office work was simulated. The results showed a significant reduction in physiological indicators such as mean body skin temperature, mean torso skin temperature, and heart rate for the three AVG scenarios, as compared to the CON condition.
I have no special remarks. Obviously, the paper was reviewed and new information was added. I suggest the paper to be accepted in its present form.
Author Response
In general, very interesting work is presented. The aim of this research was to create a cooling performance assessment for AVGs that utilize smaller fan panels. To achieve this goal, three AVGs were developed, each equipped with smaller yet more numerous ventilation fans. These garments were labelled as FFV, BBV, and FBV, with each having ten small fans on either the front, back, or both sides of the garment, respectively. As a reference point, a garment without ventilation fans but made from the same material and structure was also created (CON). Subject trials were conducted in a moderately hot environment, where 70 minutes of sedentary office work was simulated. The results showed a significant reduction in physiological indicators such as mean body skin temperature, mean torso skin temperature, and heart rate for the three AVG scenarios, as compared to the CON condition.
I have no special remarks. Obviously, the paper was reviewed and new information was added. I suggest the paper to be accepted in its present form.
Author's reply: Dear reviewer, Thank you very much for your review and comments.
Reviewer 3 Report (New Reviewer)
The article has a certain academic level and literature infrastructure. Thermal comfort is an important issue for textiles. However, I do not think that the product proposed within the scope of the research is a scientifically original product that can be used in real life.
It makes no sense to cool the body with fans and to carry equipment such as batteries, fans and cables for this purpose. While the studies on textiles have been so deep in the literature and textile products with many biological, thermodynamic and chemical properties have been developed, the fan effect with such equipment is a very ordinary idea from a scientific point of view.
It is already clear that a garment with a fan will provide more cooling than a garment without a fan. However, a structure on which the person will carry the cables, battery and fans will not be desired. Textile fabrics are much more promising in this regard. For these reasons, unfortunately, I do not find this study scientifically sufficient.
Author Response
The article has a certain academic level and literature infrastructure. Thermal comfort is an important issue for textiles. However, I do not think that the product proposed within the scope of the research is a scientifically original product that can be used in real life.
Author's Reply: Thank you very much for your precious comment. Cooling garments are used depending on the type of work, environmental conditions, etc. Previous studies In field studies showed that outdoor workers appreciated the ventilation clothing when heat stress is high (air temperature is about 37 C, humidity about 35%, this was introduced in the introduction section).
It makes no sense to cool the body with fans and to carry equipment such as batteries, fans and cables for this purpose. While the studies on textiles have been so deep in the literature and textile products with many biological, thermodynamic and chemical properties have been developed, the fan effect with such equipment is a very ordinary idea from a scientific point of view.
Author's Reply: Thank you very much for the comment. Textile fabrics alone can probably not achieve similar magnitude of cooling power of the enhanced convective and evaporative cooling by fans. For a longer period such as 8 hours work shift in hot environments, cooling garments with integrated fans is needed.
It is already clear that a garment with a fan will provide more cooling than a garment without a fan. However, a structure on which the person will carry the cables, battery and fans will not be desired. Textile fabrics are much more promising in this regard. For these reasons, unfortunately, I do not find this study scientifically sufficient.
Author's Reply: Thank you for the comment. As the above reply, textile fabrics alone can probably not achieve similar magnitude of cooling power of the enhanced evaporative and convective cooling by fans. Cooling garments with small fans are portable and save energy, avoid cooling whole buildings compared with air conditioners.
Reviewer 4 Report (New Reviewer)
I enjoyed reading this manuscript and have no doubt that it will interest the scientific community, but the following issues must be resolved the authors:
1. Repeating part of the title words as keywords is a way of wasting keywords. All title terms will act as keywords, anyway authors should come up with other terms as Keywords that people may use to find this manuscript after publication.
2. There should be Materials as sub-section 2.1, where only list of materials and equipments and their sources should be placed.
3. In line 250, authors should change each other to one another.
4. The AVGs compared with CON by the authors are FFV, BBV and FBV. In each of the conditions studied, authors need to state the comprehensive reason(s) why one of the AVGs performed more than others.
The English is fine, only minor editing is required.
Author Response
I enjoyed reading this manuscript and have no doubt that it will interest the scientific community, but the following issues must be resolved the authors:
1. Repeating part of the title words as keywords is a way of wasting keywords. All title terms will act as keywords, anyway authors should come up with other terms as Keywords that people may use to find this manuscript after publication.
Author's reply: Dear reviewer, thank you very much for your review and precious comments. The keywords have been changed according to your suggestion.
2. There should be Materials as sub-section 2.1, where only list of materials and equipments and their sources should be placed.
Author's reply: Thank you for the comment. We added sub-section in "Materials and Methods“ section according to your suggestion.
3. In line 250, authors should change each other to one another.
Author's reply: "each other" has been changed to "one another". Thank you!
4. The AVGs compared with CON by the authors are FFV, BBV and FBV. In each of the conditions studied, authors need to state the comprehensive reason(s) why one of the AVGs performed more than others.
Author's reply: Thank you for the precious comment. We have added explanation why one of the AVGs performed more than others. Please see the revised manuscript marked in red color.
Round 2
Reviewer 3 Report (New Reviewer)
Unfortunately, I still disagree. Although the developed product seems advantageous in terms of cooling the body, I do not find the combination of fan, electrical parts and textile product correct. Therefore, my opinion is negative.
Author Response
Unfortunately, I still disagree. Although the developed product seems advantageous in terms of cooling the body, I do not find the combination of fan, electrical parts and textile product correct. Therefore, my opinion is negative.
Author’s reply: Dear reviewer, thank you very much for you review and comment.
Although textile fabric plays an important role in human thermal comfort, it alone cannot cool the body when the heat stress is great and when the work shift is long. Therefore, cooling garments equipped with small fans, PCMs (phase change materials) and cooled water in pipes, etc., are needed to provide extra cooling sources. The small fans in the air ventilation garments can circulate the ambient air in the microclimate between the human body and the clothing, and in this way heat dissipation by sweat evaporation and convection is enhanced. What’ more, compared with air conditioners, the ventilation garments are portable and light weighted, avoid cooling the whole buildings, and save energy. Therefore, air ventilation garments are needed and also reported effective in laboratory and field studies (please see the Introduction section and also the following publications of the authors and other researchers). The combination of the small fan panels, electrical wires and textiles presented is the further research of the authors. The authors have carried out studies in this filed for several years and have made improvement of the air ventilation garments to provide better cooling effect and thermal comfort (please see the following the publications and the attached file of some of the authors' publications).
Author’s publications:
1. Mengmeng Zhao, Jie Yang, Faming Wang, Udayraj, Wing Ching Chan. The cooling performance of forced air ventilation garments in a warm environment: the effect of clothing eyelet designs, The Journal of The Textile Institute, 2023, 114(3): 378-387.
2. Mengmeng Zhao, Faming Wang, Chuansi Gao, Zhaoli Wang, Jun Li. The Effect of flow rate of a short sleeve air ventilation garment on torso thermal comfort in a moderate environment, Fibers and Polymers, 2021, 23(2): 546-553.
3. Mengmeng Zhao, Kalev Kuklane, Karin Lundgren, Chuansi Gao, Faming Wang. A ventilation cooling shirt worn at office work in a hot climate: cool or not? International Journal of Occupational Safety and Ergonomics, 2015, 21: 457-463.
4. Mengmeng Zhao, Chuansi Gao, Jun Li, Faming Wang. Effects of two cooling garments on post-exercise thermal comfort of female subjects in the heat. Fibers and Polymers, 2015, 16(6): 1403-1409.
5. Mengmeng Zhao, Chuansi Gao, Faming Wang, Kalev Kuklane, Ingvar Holmér, Jun Li. A Study on Local Cooling of Garments with Ventilation Fans and Openings Placed at Different Torso Sites. International Journal of Industrial Ergonomics. 2013, 43(3): 232-237.
6. Leonidas G. Ioannou, Konstantinos Mantzios, Lydia Tsoutsoubi, Eleni Nintou, Maria Vliora, Paraskevi Gkiata, Constantinos N. Dallas, Giorgos Gkikas, Gerasimos Agaliotis, Kostas Sfakianakis, Areti K. Kapnia, Davide J. Testa, Tânia Amorim, Petros C. Dinas, Tiago S. Mayor, Chuansi Gao, Lars Nybo and Andreas D. Flouris. Occupational Heat Stress: Multi-Country Observations and Interventions. International Journal of Environmental Research and Public Health, 2021, 18, 6303. https://doi.org/10.3390/ijerph18126303.
7. Chuansi Gao, Lars Nybo, Alessandro Messeri, Marco Morabito, Li-Yen Lin, Faming Wang, Kalev Kuklane, Andreas Flouris. Clothing and occupational heat stress across European industries in hot climates. 8th European Conference on Protective Clothing, 7th to 9th of May 2018, Porto, Portugal, P. 42-44.
8. Chuansi Gao, Kalev Kuklane, Per-Olof Östergren, Tord Kjellstrom. Occupational heat stress assessment and protective strategies in the context of climate change. International Journal of Biometeorology. 2018, 62: 359–371.
9. Ferraro S, Falcone T, Morabito M, Messeri A, Bonafede M, Marinaccio A, Gao C, and Molinaro V. A potential wearable solution for preventing heat strain in workplaces: The cooling effect and the total evaporative resistance of a ventilation jacket. Environmental Research, 2022, 212: 113475.
Other researcher’s publications:
1.Sun Y, and Jasper WJ. Numerical modeling of heat and moisture transfer in a wearable convective cooling system for human comfort. Building and Environment, 2015, 93: 50-62.
2. Choudhary B, Udayraj, Wang F, Ke Y, and Yang J. Development and experimental validation of a numerical model based on CFD of the human torso wearing air ventilation clothing. International Journal of Heat and Mass Transfer, 2020, 147: 118973.
3. Yi W, Zhao Y, and Chan APC. Evaluation of the ventilation unit for personal cooling system (PCS). International Journal of Industrial Ergonomics, 2017, 58: 62-68.
4. Yang J, Wang F, Song G, Li R, and Udayraj. Effects of clothing size and air ventilation rate on cooling performance of air ventilation clothing in a warm condition. International Journal of Occupational Safety and Ergonomics (JOSE), 2022, 28(1): 354-363.

Round 3
Reviewer 3 Report (New Reviewer)
The article has been corrected in detail regarding the referee suggestions. Considering the authors' previous studies on this subject, other reviewers' evaluations and the pasion of the authors for their paper in this field, I think that the paper is appropriate to accept .
This manuscript is a resubmission of an earlier submission. The following is a list of the peer review reports and author responses from that submission.
Round 1
Reviewer 1 Report
The article written by Zhao et al., titled "Development of Air Ventilation Garments with Small Fan Panels to 3 Improve Thermal Comfort" studied the use of lightweight AVGs to provide comfort against heat for workers. The article is well-written and the data is well-presented. I would like to include the following minor comments-
1. I would like to suggest that the authors should report the age of the test subjects and include it as a factor in analyzing the reported data. Also what criteria are taken into consideration before selecting the test subjects?